# Learning Sparse Approximate Inverse Preconditioners for Conjugate Gradient Solvers on GPUs

**Zherui Yang**[1]    **Zhehao Li**[2]    **Kangbo Lyu**[3,4]    **Yixuan Li**[1]    **Tao Du**[3,4]    **Ligang Liu**[1§]

[1]University of Science and Technology of China    [2]Stanford University
[3]Tsinghua University    [4]Shanghai Qi Zhi Institute

## Abstract

The conjugate gradient solver (CG) is a prevalent method for solving symmetric and positive definite linear systems $\mathbf{Ax} = \mathbf{b}$, where effective preconditioners are crucial for fast convergence. Traditional preconditioners rely on prescribed algorithms to offer rigorous theoretical guarantees, while limiting their ability to exploit optimization from data. Existing learning-based methods often utilize Graph Neural Networks (GNNs) to improve the performance and speed up the construction. However, their reliance on incomplete factorization leads to significant challenges: the associated triangular solve hinders GPU parallelization in practice, and introduces long-range dependencies which are difficult for GNNs to model. To address these issues, we propose a learning-based method to generate GPU-friendly preconditioners, particularly using GNNs to construct Sparse Approximate Inverse (SPAI) preconditioners, which avoids triangular solves and requires only two matrix-vector products at each CG step. The locality of matrix-vector product is compatible with the local propagation mechanism of GNNs. The flexibility of GNNs also allows our approach to be applied in a wide range of scenarios. Furthermore, we introduce a statistics-based scale-invariant loss function. Its design matches CG's property that the convergence rate depends on the condition number, rather than the absolute scale of $\mathbf{A}$, leading to improved performance of the learned preconditioner. Evaluations on three PDE-derived datasets and one synthetic dataset demonstrate that our method outperforms standard preconditioners (Diagonal, IC, and traditional SPAI) and previous learning-based preconditioners on GPUs. We reduce solution time on GPUs by 40%-53% (68%-113% faster), along with better condition numbers and superior generalization performance.

## 1 Introduction

Solving symmetric positive definite(SPD) sparse linear systems $\mathbf{Ax} = \mathbf{b}$ is essential in scientific computing and has wide applications in science and engineering, while the development of efficient numerical solvers remains a challenge. The conjugate gradient (CG) method is widely used for its efficiency, yet the convergence rate depends critically on preconditioning techniques that effectively transform the original linear system into an easier one to solve at low computational cost.

Traditional preconditioners have a long history of development and have achieved considerable success. Although modern GPUs provide massive parallelism, existing preconditioners struggle to leverage it, becoming the computational bottleneck despite their convergence benefits. The diagonal preconditioner offers a simple and GPU-friendly implementation but provides limited convergence improvement. Incomplete Cholesky (IC) demonstrates strong CPU performance and effectiveness in ill-conditioned problems [1]. However, its requires two triangular solves at each CG step, which are typically difficult to parallelize [2, 3], hindering its performance on GPUs.

---

[§]Corresponding author: `lgliu@ustc.edu.cn`

Recent learning-based approaches [4–6] aim to improve traditional approaches using neural networks, particularly factorization-based methods (e.g., IC) and graph neural networks (GNNs). By substituting the sequential factorization process with a GNN evaluation, these methods reduce the computational time required for incomplete factorizations. The trainability of GNNs enables the generation of higher-quality factorizations, thereby accelerating the convergence of iterative solvers. However, these methods also come with certain limitations. Prior works inherit the limitation of factorization-based methods that involve triangular solves, restricting their GPU parallelization efficiency. The triangular solves also pose challenges for GNNs to capture long-range dependencies and global information through the elimination tree, while GNNs aggregate information from local neighborhoods and struggle to effectively model such interactions [7]. Additionally, many existing approaches require computing the solution vector $\mathbf{x} = \mathbf{A}^{-1}\mathbf{b}$ of the original linear system for every $\mathbf{A}$ in the dataset to evaluate their loss functions [4, 8, 9]. Generating datasets containing large matrices becomes increasingly computationally expensive. The varying scales of $\mathbf{A}$ further pose difficulties for models in learning to improve the condition number of the preconditioned system.

To address these challenges, we present a novel approach for learning GPU-friendly preconditioners using GNNs. Our approach directly approximates the inverse $\mathbf{A}^{-1}$ with a sparse matrix and formulates its construction as a graph-learning problem, which shares conceptual parallels with the traditional Sparse Approximate Inverse (SPAI) preconditioner [10, 11]. The sparse matrix-vector product (SpMV) is the only routine required at each CG iteration, allowing our approach to leverage GPU acceleration throughout both the construction and CG solving phases. We argue that the local nature of SpMV naturally complements the local propagation mechanism of GNNs. Our approach avoids imposing restrictive triangular structures on the preconditioner, enabling us to preserve the sparsity pattern of $\mathbf{A}$ or even use a different one. This design results in improved condition numbers compared to previous SPAI and factorization-based preconditioners. Furthermore, to eliminate dependence on solution vectors and enable robust training across various datasets, we propose the Scale invariant Aligned Identity Loss (SAI loss). This loss requires only the input matrix $\mathbf{A}$, and its inherent $\mathbf{A}$-scale invariance aligns with the scale-invariant convergence behavior of CG solvers, yielding higher performance of learned preconditioners. Extensive experiments conducted on three PDE-derived datasets and a synthetic dataset validate the effectiveness of our approach compared to previous works. The results show that our approach achieves up to 113% speedup compared with standard traditional preconditioning techniques and previous learning-based preconditioners. Our approach also achieves a better condition number while maintaining good generalization performance.

In summary, we make the following contributions:

1. We propose a learning-based approach to generating GPU-friendly preconditioners, leveraging the GPU parallelism across both the construction and CG solving stages. Our approach focuses on generating the SPAI preconditioner using GNNs, exploiting the natural alignment between the local computation of SpMV and the local propagation mechanism of GNNs.

2. We introduce the SAI loss, a novel statistics-based and $\mathbf{A}$-scale invariant loss function, which reduces datasets' generation costs and enhances the performance of learned preconditioners.

3. Extensive experiments on three PDE-derived datasets and a synthetic dataset demonstrate that our method achieves lower condition numbers and reduced computation time compared to existing approaches, while also maintaining generalizability, scalability, and robustness.

## 2 Related Works

**Traditional Preconditioners**  Preconditioners $\mathbf{M}^{-1}$ transform the original linear system $\mathbf{Ax} = \mathbf{b}$ into one easier to solve $\mathbf{M}^{-1}\mathbf{A} = \mathbf{M}^{-1}\mathbf{b}$ for iterative solvers at low computational cost. The diagonal preconditioner scales the original system by its diagonal entries [11], offering a straightforward and GPU-friendly implementation. However, it provides limited improvement to the convergence rate. Incomplete Cholesky (IC) preconditioner [1] computes sparse and incomplete triangular factors ($\mathbf{M} = \mathbf{LL}^{\top} \approx \mathbf{A}$). It offers better spectral approximations but suffers from limited parallelism due to its reliance on triangular solves [12]. Sparse Approximate Inverse (SPAI) preconditioners [13–15] construct a sparse matrix $\mathbf{M}^{-1} = \mathbf{GG}^{\top} \approx \mathbf{A}^{-1}$ under sparsity constraints. At each CG step, SPAI use matrix-vector products instead of triangular solves in IC, enabling parallelism suitable for GPU architectures. However, its construction depends on sequential algorithms and often exhibits suboptimal performance. Alternative techniques, such as polynomial [11], algebraic multigrid (AMG)

[16], and domain decomposition [17] preconditioners, each balance robustness, memory usage, and parallel efficiency differently. While effective in their respective domains, they lack generality and often require careful parameter tuning to ensure performance [18, 19]. Machine learning offers a promising solution to address this limitation.

**Learning-based Preconditioners**  Recent research in learning-based preconditioning has progressed across several directions. CNNs [20], GNNs [21] and neural operators [8], which directly approximate the action of matrix inversion $\text{NN}(\mathbf{A}, \mathbf{b}) \approx \mathbf{A}^{-1}\mathbf{b}$, are utilized as preconditioners in many iterative solvers. Although these approaches can significantly reduce the number of iterations, they involve a full forward pass through the network at each iteration, thereby increasing computational overhead. The auxiliary components in neural operators are employed as a subspace in iterative solvers [22–24]. The usage of neural operators restricts their applicability primarily to PDE-related problems, while neglecting more general problems. Li et al. [4], Häusner et al. [5], Trifonov et al. [6] have investigated the integration of machine learning and factorization-based preconditioners, specifically by training GNNs to predict the factor $\mathbf{L}$ in IC preconditioners. However, these methods suffer from the triangular solve due to their sequential computation [10] and the theoretical challenges for GNNs to model long-range dependencies along the elimination tree [7]. Although these methods have achieved performance gains compared with the traditional alternatives, few of them consider the performance on GPUs, while our approach focuses specifically on GPU performance.

# 3 Our Approach

## 3.1 Problem Setup

We consider solving the sparse and SPD linear system $\mathbf{A}\mathbf{x} = \mathbf{b}$, where $\mathbf{A} \in \mathbb{R}^{n \times n}$ (or $\mathbb{R}^{nb \times nb}$ for a blocked sparse matrix with $b$ as block size[1]) arises from diverse applications, such as spatial discretization of PDEs using the finite element method (FEM). The matrix $\mathbf{A}$ is associated with a graph $\mathcal{G}_{\mathbf{A}} = (\mathcal{V}, \mathcal{E})$, defined as follows:

1. Vertices $\mathcal{V}$: Represent variables (e.g., mesh nodes in FEM) with features $\mathbf{v}_i \in \mathbb{R}^d$, that encode geometric or physical properties.
2. Edges $\mathcal{E} = \{(i, j) \mid \mathbf{A}_{ij} \neq \mathbf{0}\}$: Connect interacting variables, with edge features $\mathbf{e}_{ij}$ representing the corresponding entries or blocks $\mathbf{A}_{ij}$.

To accelerate CG's convergence, a preconditioner $\mathbf{M}^{-1}$ is introduced, which transforms the problem into an easier one to solve by modifying the search direction. The residual vector $\mathbf{r}$ is replaced with the preconditioned direction $\mathbf{s} = \mathbf{M}^{-1}\mathbf{r}$ at each CG step, as shown in Section A. *A good preconditioner* $\mathbf{M}^{-1}$ *reduces the condition number* $\kappa$ *of the linear system by effectively approximating* $\mathbf{A}^{-1}$ and a lower condition number typically indicates faster CG convergence. We consider the following preconditioner construction task based on the graph and its associated features:

$$(\mathcal{G}_{\mathbf{A}}, \{\mathbf{v}_i\}, \{\mathbf{e}_{ij}\}) \xrightarrow{\text{Algorithm}} \mathbf{M}^{-1} \in \mathbb{R}^{n \times n}, \tag{1}$$

where "Algorithm" denotes both traditional approaches (e.g., IC, SPAI, AMG) and neural networks. Traditional approaches construct $\mathbf{M}^{-1}$ through prescribed and heuristic rules based on the graph structure and its features, while the learning-based approaches infer the mapping from data.

## 3.2 Graph Neural Networks for SPAI construction

### 3.2.1 Sparse Approximate Inverse Preconditioners (SPAI)

To ensure high computational efficiency, particularly on GPUs, we directly employ a sparse approximate inverse $\mathbf{M}^{-1}$ as the preconditioner for $\mathbf{A}^{-1}$ (SPAI). Additionally, since the convergence of CG depends on a symmetric and positive definite $\mathbf{M}^{-1}$, we propose to factorize the preconditioner as

$$\mathbf{A}^{-1} \approx \mathbf{M}^{-1} = \mathbf{G}\mathbf{G}^{\top} + \varepsilon\mathbf{I}, \tag{2}$$

where the sparse matrix $\mathbf{G}$ is the output of the GNN and $\varepsilon$ is a small positive constant to enforce SPD property, as shown in Figure 1. Numerous prior studies [15, 14, 13] have demonstrated that sparse

---

[1]In the hyperelasticity problem, there are 3 variables to solve per node, resulting in a block size of 3.

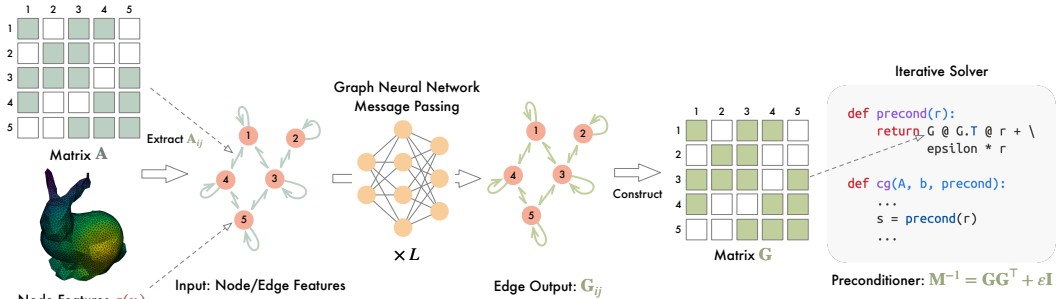

Figure 1: Overview of our approach: By inputting the matrix's nonzero entries $\mathbf{A}_{ij}$ and node features $a(x_i)$, the GNN processes these features through message passing, and outputs the entries of $\mathbf{G}_{ij}$. The sparse matrix $\mathbf{G}$ is assembled and then applied in the preconditioned CG solver.

$\mathbf{M}^{-1}$ can effectively reduce the condition number, even when $\mathbf{A}^{-1}$ is dense. As a result, at each CG iteration, the application of preconditioner $\mathbf{s} = \mathbf{M}^{-1}\mathbf{r}$ is instantiated as two sparse matrix-vector products and a vector addition, which is both efficient on GPUs.

Previous works [4, 5] take an opposite approach to ours. Their approaches output a sparse triangular matrix $\mathbf{L}$, which aim to approximate $\mathbf{A}$ with $\mathbf{L}\mathbf{L}^\top$, resulting in Incomplete Cholesky (IC) precon­ditioner $\mathbf{M}^{-1} = (\mathbf{L}\mathbf{L}^\top)^{-1}$. The application of preconditioner $\mathbf{L}\mathbf{L}^\top\mathbf{s} = \mathbf{r}$ involves two triangular solves: forward substitution $\mathbf{L}\mathbf{y} = \mathbf{r}$ and backward substitution $\mathbf{L}^\top\mathbf{s} = \mathbf{y}$. Although triangular solves can be efficient on CPUs, the inherent sequential nature limits their performance on GPUs.

**Locality of SPAI Preconditioner**  Consider the application of our preconditioner $\mathbf{s} = \mathbf{M}^{-1}\mathbf{r}$. Suppose $\mathbf{G}$ shares the same sparsity pattern as $\mathbf{A}$, the $j$-th entry of the output vector $\mathbf{s}$ is

$$\mathbf{s}_j = (\mathbf{M}^{-1}\mathbf{r})_j = \varepsilon\mathbf{r}_j + \sum_{\substack{l \\ \mathbf{A}_{jl} \neq 0}} \mathbf{G}_{jl} \sum_{\substack{k \\ \mathbf{A}_{kl} \neq 0}} \mathbf{G}_{kl}\mathbf{r}_k. \tag{3}$$

Since the non-zero entries of $\mathbf{A}$ and the entries of vectors correspond to edges and nodes in the graph, Equation (3) reveals that *the output on node $j$ depends only on its two-hop neighborhood*, mediated through intermediate nodes $l$ and source nodes $k$. This makes SPAI fundamentally more compatible with GNN architectures than factorization-based methods.

In contrast, in IC preconditioner, the forward substitution $\mathbf{L}\mathbf{y} = \mathbf{r}$ and backward substitution $\mathbf{L}^\top\mathbf{s} = \mathbf{y}$ propagate information along the elimination tree induced by a chosen node ordering. The forward substitution aggregates values from ancestor nodes, while backward substitution collects from descendant nodes, resulting in global dependencies. This hierarchical propagation in IC conflicts with the local aggregation mechanism of GNNs over fixed-hop neighborhoods. Furthermore, IC requires explicit directional dependencies (enforcing lower triangular $\mathbf{L}$), which are inconsistent with the undirected/symmetric assumptions in existing GNN-based preconditioners.

### 3.2.2   GNN Architecture

Our GNN follows an encoder-processor-decoder architecture, adapted from Gilmer et al. [25]. Given node features $\mathbf{v}_i \in \mathbb{R}^{d_{\text{node}}}$ and edge features $\mathbf{e}_{ij} \in \mathbb{R}^{d_{\text{edge}}}$, the GNN first applies two MLPs $E_n, E_e$ to encode the node and edge features into their hidden representations $\mathbf{x}^{(0)}, \mathbf{h}^{(0)} \in \mathbb{R}^d$:

$$\mathbf{x}_i^{(0)} = E_n(\mathbf{v}_i), \quad \mathbf{h}_{ij}^{(0)} = E_e(\mathbf{e}_{ij}). \tag{4}$$

Subsequently, $L$ message-passing layers are applied. Each layer utilizes three MLPs $f_m^{(t)}, f_v^{(t)}, f_e^{(t)}$, which serve as message functions and update functions for node and edge features:

$$\mathbf{m}^{(t)} = \sum_{j \in N(i)} f_m^{(t)}(\mathbf{x}_i^{(t-1)}, \mathbf{x}_j^{(t-1)}, \mathbf{e}_{ij}),$$

$$\mathbf{x}_i^{(t)} = \mathbf{x}_i^{(t-1)} + f_v^{(t)}(\mathbf{m}_i^{(t)}), \quad \mathbf{h}_{ij}^{(t)} = \mathbf{h}_{ij}^{(t-1)} + f_e^{(t)}(\mathbf{x}_i^{(t)}, \mathbf{x}_j^{(t)}, \mathbf{h}_{ij}^{(t-1)}). \tag{5}$$

Finally, a decoder MLP $D$ is applied to the edge features of the last layer $\mathbf{h}^{(L)}$ to get the block entries:

$$\mathbf{G}_{ij} = D(\mathbf{h}_{ij}^{(L)}) \in \mathbb{R}^{b \times b}. \tag{6}$$

The global sparse matrix $\mathbf{G} \in \mathbb{R}^{nb \times nb}$ is constructed by replacing each block $\mathbf{A}_{ij}$ with $\mathbf{G}_{ij}$ while preserving the original sparsity pattern.

### 3.3  Loss Function

The loss function is crucial for learning effective preconditioners $\mathbf{M}$. While traditional SPAI approaches aim to minimize $\|\mathbf{M}^{-1} - \mathbf{A}^{-1}\|_F$ or $\|\mathbf{A}\mathbf{M}^{-1} - \mathbf{I}\|_F$, these formulations suffer from two limitations: (1) Direct evaluation of $\mathbf{A}\mathbf{M}^{-1}$ requires significantly larger memory and computation resources for large matrices even when both $\mathbf{A}$ and $\mathbf{M}$ are sparse. (2) These losses depend on the absolute magnitude of $\mathbf{A}$, while the convergence rate of CG solvers is invariant to the magnitude.

**Stochastic Estimation**  To address the computational bottleneck, we first adopt the stochastic trace estimator [5, 26] to approximate the matrix Frobenius norm without explicitly constructing $\mathbf{A}\mathbf{M}^{-1}$:

$$\|\mathbf{A}\mathbf{M}^{-1} - \mathbf{I}\|_F^2 = \mathrm{tr}((\mathbf{A}\mathbf{M}^{-1} - \mathbf{I})^\top (\mathbf{A}\mathbf{M}^{-1} - \mathbf{I})) = \mathbb{E}_{\mathbf{w}}\left[\|\mathbf{A}\mathbf{M}^{-1}\mathbf{w} - \mathbf{w}\|_2^2\right], \tag{7}$$

where $\mathbf{w}$ is a vector of independent and identically distributed (i.i.d.) random variables drawn from the standard normal distribution $\mathcal{N}(0, 1)$. This reduces the computation to matrix-vector products $\mathbf{A}\mathbf{M}^{-1}\mathbf{w}$, avoiding costly matrix-matrix products. The estimator can also be applied to $\|\mathbf{M}^{-1} - \mathbf{A}^{-1}\|_F^2 \approx \|\mathbf{M}^{-1}\mathbf{w} - \mathbf{A}^{-1}\mathbf{w}\|_2^2$. However, this either requires access to $\mathbf{A}^{-1}$ or involves solving $\mathbf{x} = \mathbf{A}^{-1}\mathbf{w}$ for each sample in the dataset, making it computationally expensive .

**Scale Invariance**  The loss function in (7) exhibits scale sensitivity because it depends on the absolute magnitude of $\mathbf{A}$. To align with the CG solver's scale invariance property, we normalize $\mathbf{A}$ by its norm $\|\mathbf{A}\|$ in the loss function:

$$\mathcal{L}_{\mathrm{SAI}}(\mathbf{A}, \mathbf{M}^{-1}) = \left\|\frac{1}{\|\mathbf{A}\|}\mathbf{A}\mathbf{M}^{-1} - \mathbf{I}\right\|_F^2, \quad \mathcal{L}_{\mathrm{SAI}}(\mathbf{A}, \mathbf{M}^{-1}, \mathbf{w}) = \left\|\left(\frac{1}{\|\mathbf{A}\|}\mathbf{A}\mathbf{M}^{-1} - \mathbf{I}\right)\mathbf{w}\right\|_2^2. \tag{8}$$

This formulation ensures $\mathcal{L}_{\mathrm{SAI}}(\mathbf{A}, \mathbf{M}^{-1}, \mathbf{w}) = \mathcal{L}_{\mathrm{SAI}}(\alpha\mathbf{A}, \mathbf{M}^{-1}, \mathbf{w})$ for any $\alpha > 0$. We refer to this as **Scale invariant Aligned Identity (SAI) loss**, which explicitly decouples the preconditioner learning from the matrix's absolute scale, while enforcing alignment between the scaled preconditioned matrix and the identity matrix. The condition number $\kappa$ of $\mathbf{A}\mathbf{M}^{-1}$ can be estimated as

$$\kappa(\mathbf{A}\mathbf{M}^{-1}) = \frac{\sigma_{\max}(\mathbf{A}\mathbf{M}^{-1})}{\sigma_{\min}(\mathbf{A}\mathbf{M}^{-1})} = \frac{\sigma_{\max}(\mathbf{I} + \mathbf{E})}{\sigma_{\min}(\mathbf{I} + \mathbf{E})} \leq \frac{1 + \sigma_{\max}(\mathbf{E})}{1 - \sigma_{\max}(\mathbf{E})} \approx 1 + 2\sigma_{\max}(\mathbf{E}) = 1 + 2\|\mathbf{E}\|_2, \tag{9}$$

where $\mathbf{E} = \mathbf{A}\mathbf{M}^{-1}/\|\mathbf{A}\| - \mathbf{I}$ is the error matrix and $\sigma$ denotes matrix singular values (see Section B for its detailed proof). This derivation shows that the SAI loss encourages $\mathbf{M}^{-1}$ to focus on improving $\mathbf{A}\mathbf{M}^{-1}$'s spectral properties, while being invariant to absolute magnitude of $\mathbf{A}$. Although conventional matrix norms of $\mathbf{A}$ like the Frobenius norm are theoretically valid, their scale is often sensitive to matrix dimensions and outlier entries. We therefore define a more robust and dimension-agnostic norm as $\|\mathbf{A}\| = \mathrm{mean}_{\mathbf{A}_{ij} \neq 0}|\mathbf{A}_{ij}|$. This setting also ensures that the edge features of the GNN and the loss term remain on a reasonable scale, promoting more stable and efficient optimization across matrices of varying sizes.

## 4  Experimental results

We answer the following questions through our experiments: (1) How does our approach compare with traditional and learning-based approaches? (2) Is our loss more effective than those in prior works? (3) How do our approach generalize to unseen examples? (4) Does our approach yield a better condition number compared to previous methods? We describe the experiment setup in Sec. 4.1 and provide answers to these questions in Sec. 4.2 to 4.5.

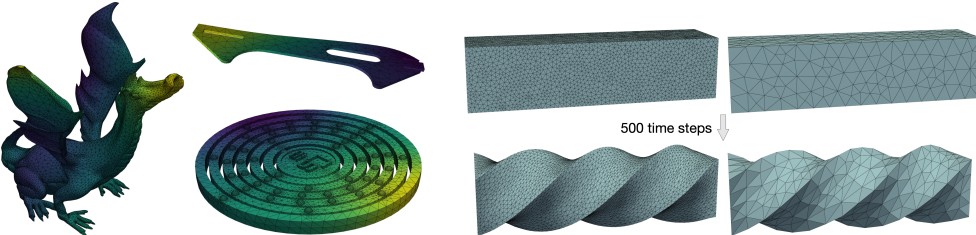

(a) Density plot of examples in TetWild Dataset for Heat/Poisson Equation test case.

(b) Starting and ending timestep of meshes with different resolution in the hyperelasticity case.

Figure 2: Examples in our PDE-derived test cases.

## 4.1 Experiment Setup

Our experiments include three PDE-derived datasets, as well as a fully algebraic dataset composed of randomly generated matrices. As shown in Figure 2, we use FEM and tetrahedron mesh to discretize the PDEs, yielding the sparse system matrix $\mathbf{A}$. The test cases considered in our study are as follows:

1. **Heat Equation**: $a(x) \cdot u_t - \Delta u = 0$, where $a(x)$ represents the spatially varying density (randomly sampled on the mesh, with values ranging from $1e-4$ to $5e-4$).

2. **Poisson Equation**: $-\Delta u = f$ with $u|_D = g$, where $D$ denotes a randomly specified Dirichlet boundary.

3. **Hyperelasticity**: $\mathbf{M\ddot{u}} + \mathbf{f}_{\text{int}}(\mathbf{u}) = \mathbf{f}_{\text{ext}}$, where $\mathbf{M}$ is the mass matrix, and $\mathbf{f}_{\text{int}}, \mathbf{f}_{\text{ext}}$ represent internal and external forces, respectively. The internal force $\mathbf{f}_{\text{int}}$ is nonlinear and derived from a stable Neo-Hookean material model [27].

4. **Synthetic System**: $\mathbf{A} = \mathbf{PP}^\top + \varepsilon\mathbf{I}$, where $\mathbf{P}$ is a random matrix and $\varepsilon = 10^{-4}$ ensures the numerical positive definiteness of $\mathbf{A}$.

For Heat and Poisson problems, we use 9,147 meshes with node counts ranging from 400 to 32,000 from the TetWild dataset [28]. All meshes are normalized to ensure the entire geometry fits within the $[-1, 1]^3$ domain. The hyperelasticity simulations involve a beam twist scene for 500 timesteps. This experiment shares the same geometry but varies in resolution and topology through remeshing with node counts ranging from 645 to 14,039 (matrix size is $3\times$ larger). The synthetic dataset contains 1,000 matrices with a sparsity of approximately 0.12%, with matrix sizes ranging from 10,000 to 20,000. All experiments employ a 4:1 train-test split, with all reported results evaluated on the test set. Detailed problem configurations, their matrix representations and corresponding inputs of GNNs are provided in Appendix C.2.

**GNN Implementation and Training** For all experiments in this work, we fixed the number of message passing steps $L$ to 4, the number of hidden layers in all the MLPs to 1, the number of neurons $d$ in the hidden layer to 24, and $\varepsilon = 10^{-4}$. The GNN has about 24k trainable parameters in total. All models are trained for 500 epochs using a batch size of 4 on a single NVIDIA A100 GPU, optimized with AdamW [29] and an exponentially decaying learning rate scheduler (decay rate = 0.99).

**Evaluation Metrics** The constructed preconditioner $\mathbf{M}^{-1}$ is subsequently applied in CG to solve $\mathbf{Ax} = \mathbf{b}$. Since each CG iteration requires one preconditioner application, the total solving time is governed by:

$$T_{\text{total}} = T_{\text{construct}} + k \times (T_{\text{apply}} + T_{\text{cg}}), \qquad (10)$$

where $k$ denotes iteration count, and $T_{\text{construct}}, T_{\text{apply}}$ represents the preconditioner's construction time at the beginning and application time per iteration, and $T_{\text{cg}}$ accounts for fixed operations per CG iteration (e.g., dot products, matrix-vector products). **The primary objective of preconditioning is to minimize $T_{\text{total}}$.** $T_{\text{apply}}$ and $T_{\text{construct}}$ rely on the computational complexity of its associated routines (such as SpMV and triangular solve), while preconditioners with better spectral approximation to $\mathbf{A}^{-1}$ or smaller condition number often result in lower $k$. This establishes a fundamental trade-off between solver efficiency and approximation quality in preconditioner design.

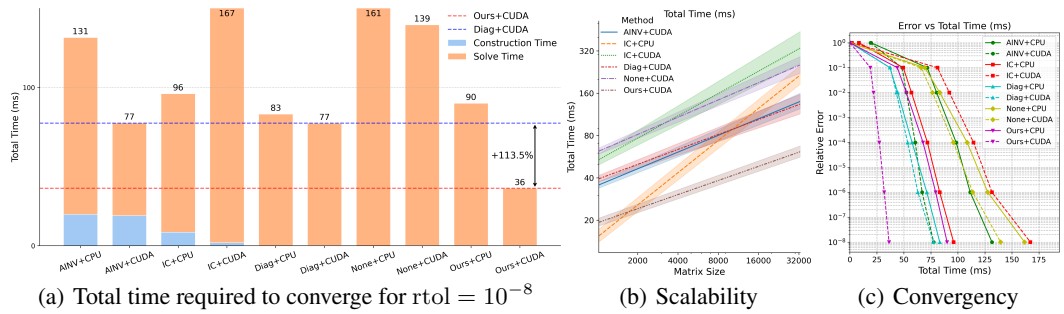

(a) Total time required to converge for rtol = $10^{-8}$ (b) Scalability (c) Convergency

Figure 3: Performance of CG with different preconditioners for the heat problem. Figure (a) compares the average total solve time $T_{\text{total}}$ and preconditioner's construction time $T_{\text{construct}}$ of CG with different preconditioners and devices. Figure (b) illustrates the relationship between matrix size and $T_{\text{total}}$, including its 95% confidence interval, demonstrating the superior scalability of our approach on GPUs. Figure (c) compares the total solve time required to achieve different rtol.

## 4.2 Preconditioner Performance

We consider three standard traditional preconditioners on both GPU and CPU architectures, including: (1) Diagonal (Diag), (2) Incomplete Cholesky (IC), and (3) Sparse Approximate Inverse (AINV)[2]. All the preconditioned CG are implemented in C++ and CUDA with OpenBLAS [30], cuBLAS, cuSPARSE [31], and cusplibrary [32] for their high-performance linear algebra kernels and preconditioner implementations. We list their implementation details in Appendix C.1.

**Comparison to Traditional Approaches**    Table 1 summarizes the time cost $T_{\text{total}}$ and iteration count $k$ for the CG method equipped with different preconditioners to converge to a relative tolerance rtol $= \|\mathbf{b} - \mathbf{Ax}\|/\|\mathbf{b}\| < 10^{-8}$. The results demonstrate that our approach outperforms all baseline approaches and achieves consistent superiority across all categories of solving tasks. Additionally, we also present a comparison of our method with AMG in Appendix D.2.

Table 1: GPU Benchmark results across different datasets. Total time $T_{\text{total}}$ (ms) and total iterations $k$ (in parentheses) until relative residual norm is less than rtol $= 10^{-8}$ are listed in the table. Lower values indicate better performance. The lowest value is in bold and the second lowest is underlined. The relative promotion (Rel. Prom.) indicates the ratio of the time saved over the second-best one.

| Test Case | Diag | IC | AINV | Ours | Rel. Prom. (↑) |
|---|---|---|---|---|---|
| Heat Equation | 77(520) | 167(204) | 78(330) | **36**(197) | 113% |
| Poisson Equation | 45(320) | 101(128) | 58(217) | **26**(128) | 73% |
| Hyperelasticity | 86(464) | 202(117) | 247(266) | **51**(175) | 68% |
| Synthetic System | 445(2775) | 1399(1808) | 5024(10896) | **253**(1122) | 75% |

**Performance Analysis Across Convergence Thresholds**    Table 2 summarizes the total solve time $T_{\text{total}}$ and iteration numbers $k$ of CG solvers using different preconditioners up to various convergence thresholds. Although the Diagonal preconditioner has the smallest construction overhead, it results in a limited improvement in convergence rate. Compared with previous preconditioners, our approach achieves the smallest $T_{\text{total}}$ to the given rtol. We also compare the scalability of our approach with that of baseline methods in Figure 3, and our approach is still efficient as the matrix size increases.

**Runtime Analysis of Routines in CG solvers**    Table 3 breaks down the runtime of each routine in CG solvers. Our approach reduces $T_{\text{total}}$ by balancing three key factors: low setup cost (comparable to Diag) $T_{\text{construct}}$, reasonable iteration counts $k$ (similar to IC), and GPU-efficient operations $T_{\text{apply}}$. This trade-off makes our approach particularly competitive in GPU settings. Although IC achieves

---

[2]SPAI is a specific subclass of AINV preconditioners that focuses on constructing a sparse approximate inverse, whereas AINV is the general parent concept of approximate inverse techniques.

comparable iteration counts $k$, its computational cost for applying the preconditioner (the triangular solves) is substantially higher than ours on GPUs, resulting in degraded performance.

Table 2: Comparison between different preconditioners for the heat problem. Total time $T_{\text{total}}$ (ms), total iterations $k$ (in parentheses), and preconditioner's construction time $T_{\text{construct}}$ (Cons.) are listed in the table. The best value is in bold, and a lower value indicates better performance.

| Stage | CPU | | | | GPU | | | |
|-------|------|------|------|------|------|------|------|------|
| | Diag | IC | AINV | Ours | Diag | IC | AINV | Ours |
| Cons. | **0.126** | 8.426 | 19.308 | 0.181 | 0.196 | 1.866 | 18.924 | 0.181 |
| $10^{-2}$ | 44(309) | 57(126) | 80(199) | 52(124) | 43(309) | 92(115) | 52(199) | **22**(124) |
| $10^{-4}$ | 58(383) | 72(157) | 98(246) | 68(154) | 53(384) | 114(143) | 61(246) | **27**(154) |
| $10^{-6}$ | 71(450) | 83(180) | 111(282) | 79(176) | 63(442) | 132(164) | 67(280) | **32**(176) |
| $10^{-8}$ | 83(511) | 96(205) | 132(328) | 90(197) | 77(520) | 167(204) | 78(330) | **36**(197) |

Table 3: Time breakdown of each routine in preconditioned CG solvers on GPUs and CPUs for the heat problem. All timings are reported in milliseconds.

| Method | CPU | | | | GPU | | | |
|--------|-----|------------------|----------------|--------------|-----|------------------|----------------|--------------|
| | $k$ | $T_{\text{construct}}$ | $T_{\text{apply}}$ | $T_{\text{cg}}$ | $k$ | $T_{\text{construct}}$ | $T_{\text{apply}}$ | $T_{\text{cg}}$ |
| Diag | 511 | 0.12 | 0.01 | | 520 | 0.12 | 0.01 | |
| IC | 205 | 8.54 | 0.26 | 0.16 | 205 | 1.88 | 0.80 | 0.14 |
| AINV | 328 | 19.0 | 0.17 | | 330 | 19.0 | 0.03 | |
| Ours | 197 | 0.18 | 0.29 | | 197 | 0.18 | 0.04 | |

**Comparison to Learning-based Approaches**   We compare our method with existing learning-based approaches on the dataset provided by previous works in Table 4. We adopt their open-source implementation with the default settings. Compared to previous learning-based methods, our approach demonstrates better performance particularly on GPUs. The matrix size provided in Li et al. [4] is sufficiently small that Diagonal preconditioner on CPUs achieves the best performance, while our approach attains the second-best performance. Compared to Häusner et al. [5], our approach achieves a lower total solve time ($T_{\text{total}}$) on GPUs.

Table 4: Comparsion to previous works. Total time $T_{\text{total}}$ (ms) and total iterations $k$ (in parentheses) until relative residual norm is less than $\text{rtol} = 10^{-8}$ are listed in the table. Prev. corresponds to the performance of previous works. The lowest value is in bold and the second lowest is underlined.

| Device | Test Case | Diag | IC | AINV | Prev. | Ours |
|--------|-----------|------|------|------|-------|------|
| CPU | Li et al. [4] | **12**(208) | 19(90) | 33(108) | 26(108) | 17(102) |
| | Häusner et al. [5] | **799**(970) | 1387(438) | 3235(753) | 1139(354) | 1320(456) |
| GPU | Li et al. [4] | 29(208) | 45(87) | 30(108) | 51(108) | **26**(102) |
| | Häusner et al. [5] | 166(970) | 440(385) | 2456(753) | 1040(354) | **132**(456) |

## 4.3   Generalizability and Robustness

We present the performance of our approach on out-of-distribution test samples in Table 5. For the heat problem, we evaluate at a fixed mesh density of 1e-3 (Heat-Density) and further assess generalization to finer meshes with more than 32k nodes (Heat-Large). For the hyperelasticity problem, we test on the same geometry with a finer mesh (22,618 nodes). For the synthetic problem, we use matrices with size ranging from 48k to 96k (Synthetic-Large). Comparing the relative promotions on each test case, our approach generalizes well to unseen resolutions and physical parameters.

Table 5: Test on out-of-distribution data on GPUs. The total time (ms) and total iterations (in parentheses) are reported. The best value is in bold, and a lower value indicates better performance. In-distribution (In) and out-of-distribution (Out) relative promotions are recorded.

| Test Case | Diag | IC | AINV | Ours | Rel. Prom. ($\uparrow$) Out | In |
|---|---|---|---|---|---|---|
| Heat-Density | 62(468) | 135(175) | 67(307) | **35**(201) | 80% | 113% |
| Heat-Large | 251(1033) | 808(388) | 407(740) | **154**(409) | 62% | 73% |
| Hyperelasticity | 326(1005) | 667(287) | 1154(496) | **236**(359) | 72% | 68% |
| Synthetic-Large | 485(1575) | 1616(781) | 19840(5591) | **347**(559) | 39% | 75% |

## 4.4 Ablation Study on Loss Functions

We compare SAI loss with cosine similarity loss $\mathcal{L}_{CS} = \mathbf{w}^\top \mathbf{AM}^{-1}\mathbf{w}/\|\mathbf{w}\|\|\mathbf{AM}^{-1}\mathbf{w}\|$ [7] and the scale variant loss $\mathcal{L}_2 = \|\mathbf{AM}^{-1}\mathbf{w} - \mathbf{w}\|_2^2$. As shown in Table 6, although $\mathcal{L}_2$ performs similarly on the heat problem, the proposed SAI loss demonstrates better effectiveness on more difficult test cases.

Table 6: Comparison of different losses. The average iteration counts are recorded. Since $T_{\text{construct}}$ and $T_{\text{apply}}$ are shared, a lower value indicates a better result and a smaller total solving time.

| Loss | Heat | Poisson | Hyperelasticity | Synthetic |
|---|---|---|---|---|
| $\mathcal{L}_2$ | **195.6** | 134.1 | 185.4 | 2109.8 |
| $\mathcal{L}_{CS}$ | 207.3 | 133.4 | 182.5 | 2185.7 |
| $\mathcal{L}_{SAI}$ | 197.4 | **128.8** | **175.7** | **1122.0** |

## 4.5 Condition Number of Preconditioned Matrix

The condition numbers provide a theoretical measure of the preconditioner's performance. Besides the standard condition number $\kappa$, we also compute the Kaporin's condition number [33], which is the ratio of the arithmetic mean to the geometric mean of the matrix eigenvalues $\lambda_i$:

$$\kappa(\mathbf{AM}^{-1}) = \frac{\max_i \lambda_i}{\min_i \lambda_i}, \quad \kappa_{\text{Kaporin}}(\mathbf{AM}^{-1}) = \frac{(\sum_{i=1}^N \lambda_i)/N}{(\lambda_1 \lambda_2 \cdots \lambda_N)^{\frac{1}{N}}}. \tag{11}$$

We evaluate the condition numbers on a simplified heat problem on a single mesh with 3,764 nodes and varying diffusivity coefficients. Figure 4 demonstrates that our method notably reduces the condition number when compared with earlier traditional approaches, revealing its efficacy.

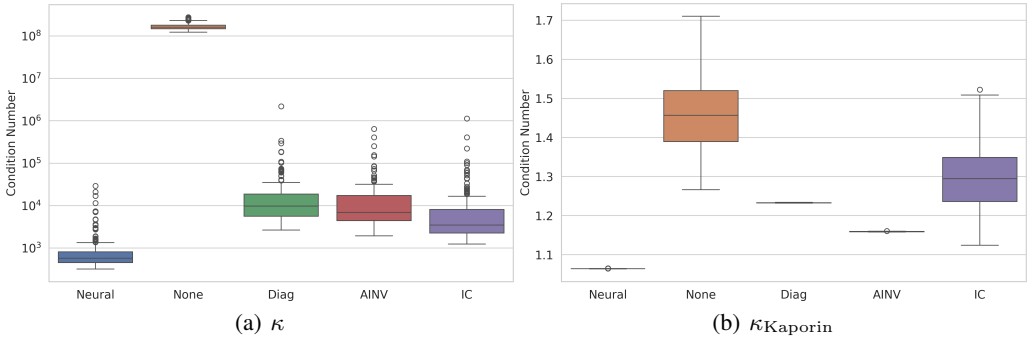

(a) $\kappa$        (b) $\kappa_{\text{Kaporin}}$

Figure 4: Condition number distributions. Median, IQR, and outliers are shown. A smaller conditioner number indicates better performance of the preconditioner, and the lower bound is 1.

# 5 Conclusion

This work proposes a learning-based approach for constructing SPAI preconditioners, aiming at accelerating the convergence of CG solvers on GPUs. Our method leverages the natural alignment between the local propagation mechanism of GNNs and the localized structure of SPAI preconditioners. Furthermore, we propose the SAI loss to reduce the computational cost of training data generation and improve the quality of the learned preconditioners. Experimental results demonstrate that our approach delivers consistent speedups over traditional and existing learning-based methods, with improved robustness, generalization, and compatibility with parallel computing architectures.

**Limitations and Future Works**    Although our approach outperforms existing methods in accelerating CG solvers, several limitations remain. First, we enforce the sparsity pattern of $\mathbf{G}$ to exactly match that of $\mathbf{A}$, while many existing approaches employ dynamic dropping strategies to further limit fill-in or incorporate two-hop connection in $\mathbf{G}$ to improve the effectiveness of preconditioner. Second, while our framework could potentially be extended to other Krylov subspace methods such as GMRES or integrated into multigrid frameworks (e.g., as a smoother), this work focuses specifically on SPD systems and the CG solver. Third, our current work is limited by the memory of a single GPU, and a promising direction for future work is to scale our approach to multi-GPU systems using techniques from distributed GNNs.

## Acknowledgments and Disclosure of Funding

This work is supported by the National Key R&D Program of China (2022YFB3303400) and the National Natural Science Foundation of China (62025207). Tao Du acknowledges the research funding support from Tsinghua University and Shanghai Qi Zhi Institute.

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

## A   Preconditioned Conjugate Gradient Algorithm

We describe the standard PCG algorithm [34] here.

## B   Proof of the Inequality

The inequality concerns the condition number $\kappa(\mathbf{A}\mathbf{M}^{-1})$. The error matrix is defined in the main text as $\mathbf{E} = \mathbf{A}\mathbf{M}^{-1}/\|\mathbf{A}\| - \mathbf{I}$. From this definition, we can write:

$$\mathbf{I} + \mathbf{E} = \frac{\mathbf{A}\mathbf{M}^{-1}}{\|\mathbf{A}\|}. \tag{12}$$

The condition number $\kappa(\mathbf{A}\mathbf{M}^{-1})$ is given by:

$$\kappa(\mathbf{A}\mathbf{M}^{-1}) = \frac{\sigma_{\max}(\mathbf{A}\mathbf{M}^{-1})}{\sigma_{\min}(\mathbf{A}\mathbf{M}^{-1})}. \tag{13}$$

Since for any non-zero scalar $c$, $\kappa(c\mathbf{X}) = \kappa(\mathbf{X})$, we have:

$$\kappa(\mathbf{A}\mathbf{M}^{-1}) = \kappa\left(\frac{\mathbf{A}\mathbf{M}^{-1}}{\|\mathbf{A}\|}\right) = \kappa(\mathbf{I} + \mathbf{E})\frac{\sigma_{\max}(\mathbf{I} + \mathbf{E})}{\sigma_{\min}(\mathbf{I} + \mathbf{E})}. \tag{14}$$

We use Weyl's inequality for singular values. For any square matrices $\mathbf{X}$ and $\mathbf{Y}$ of the same dimensions, Weyl's inequality states:

$$|\sigma_k(\mathbf{X} + \mathbf{Y}) - \sigma_k(\mathbf{X})| \le \sigma_{\max}(\mathbf{Y}), \tag{15}$$

---

**Algorithm 1** Preconditioned Conjugate Gradient

---
$i \leftarrow 0$
$\mathbf{r} \leftarrow \mathbf{b} - A\mathbf{x}$
$\mathbf{d} \leftarrow M^{-1}\mathbf{r}$                                    // Apply Preconditioner
$\delta_{\text{new}} \leftarrow \mathbf{r}^T\mathbf{d}$
$\delta_0 \leftarrow \delta_{\text{new}}$
**while** $i < i_{\max}$ **and** $\delta_{\text{new}} > \varepsilon^2\delta_0$ **do**
    $\mathbf{q} \leftarrow A\mathbf{d}$
    $\alpha \leftarrow \frac{\delta_{\text{new}}}{\mathbf{d}^T\mathbf{q}}$
    $\mathbf{x} \leftarrow \mathbf{x} + \alpha\mathbf{d}$
    **if** $i$ is divisible by 50 **then**
        $\mathbf{r} \leftarrow \mathbf{b} - A\mathbf{x}$
    **else**
        $\mathbf{r} \leftarrow \mathbf{r} - \alpha\mathbf{q}$
        $\mathbf{s} \leftarrow M^{-1}\mathbf{r}$                              // Apply Preconditioner
        $\delta_{\text{old}} \leftarrow \delta_{\text{new}}$
        $\delta_{\text{new}} \leftarrow \mathbf{r}^T\mathbf{s}$
        $\beta \leftarrow \frac{\delta_{\text{new}}}{\delta_{\text{old}}}$
        $\mathbf{d} \leftarrow \mathbf{s} + \beta\mathbf{d}$
    **end if**
    $i \leftarrow i + 1$
**end while**

---

where $\sigma_k(\cdot)$ is the $k$-th singular value and $\sigma_{\max}(\mathbf{Y})$ is the largest singular value of $\mathbf{Y}$ (i.e., its spectral norm $\|\mathbf{Y}\|_2$).

Let $\mathbf{X} = \mathbf{I}$ (the identity matrix) and $\mathbf{Y} = \mathbf{E}$. The singular values of $\mathbf{I}$ are all 1, so $\sigma_k(\mathbf{I}) = 1$ for all $k$. Applying Weyl's inequality to $\mathbf{I} + \mathbf{E}$:

$$|\sigma_k(\mathbf{I} + \mathbf{E}) - \sigma_k(\mathbf{I})| \leq \sigma_{\max}(\mathbf{E}) \tag{16}$$

$$|\sigma_k(\mathbf{I} + \mathbf{E}) - 1| \leq \sigma_{\max}(\mathbf{E}). \tag{17}$$

This implies that for any singular value $\sigma_k(\mathbf{I} + \mathbf{E})$:

$$1 - \sigma_{\max}(\mathbf{E}) \leq \sigma_k(\mathbf{I} + \mathbf{E}) \leq 1 + \sigma_{\max}(\mathbf{E}). \tag{18}$$

This inequality holds for both the maximum and minimum singular values of $\mathbf{I} + \mathbf{E}$:

$$\sigma_{\max}(\mathbf{I} + \mathbf{E}) \leq 1 + \sigma_{\max}(\mathbf{E}) \tag{19}$$

$$\sigma_{\min}(\mathbf{I} + \mathbf{E}) \geq 1 - \sigma_{\max}(\mathbf{E}). \tag{20}$$

For the lower bound on $\sigma_{\min}(\mathbf{I} + \mathbf{E})$ to be positive, and thus for $\mathbf{I} + \mathbf{E}$ to be invertible, we require $\sigma_{\max}(\mathbf{E}) < 1$. This condition is typically met when the preconditioner $\mathbf{M}$ is effective, making $\mathbf{A}\mathbf{M}^{-1}/\|\mathbf{A}\|$ close to $\mathbf{I}$.

Under the condition $\sigma_{\max}(\mathbf{E}) < 1$, we can bound the condition number:

$$\kappa(\mathbf{A}\mathbf{M}^{-1}) = \kappa(\mathbf{I} + \mathbf{E}) = \frac{\sigma_{\max}(\mathbf{I} + \mathbf{E})}{\sigma_{\min}(\mathbf{I} + \mathbf{E})} \leq \frac{1 + \sigma_{\max}(\mathbf{E})}{1 - \sigma_{\max}(\mathbf{E})}. \tag{21}$$

This establishes the first part of the inequality.

For the approximation, if $\sigma_{\max}(\mathbf{E})$ is small (i.e., $\sigma_{\max}(\mathbf{E}) \ll 1$), we can use the Taylor expansion for $(1 - x)^{-1} = 1 + x + x^2 + \ldots$ for $|x| < 1$. Let $x = \sigma_{\max}(\mathbf{E})$. Then:

$$\frac{1 + x}{1 - x} = (1 + x)(1 - x)^{-1} = (1 + x)(1 + x + x^2 + O(x^3)) = 1 + 2x + 2x^2 + O(x^3). \tag{22}$$

For $x \ll 1$, we can approximate this as:

$$\frac{1 + \sigma_{\max}(\mathbf{E})}{1 - \sigma_{\max}(\mathbf{E})} \approx 1 + 2\sigma_{\max}(\mathbf{E}). \tag{23}$$

Finally, recall that the spectral norm $\|\mathbf{E}\|_2$ is defined as $\sigma_{\max}(\mathbf{E})$. Therefore,

$$\kappa(\mathbf{A}\mathbf{M}^{-1}) \leq \frac{1 + \sigma_{\max}(\mathbf{E})}{1 - \sigma_{\max}(\mathbf{E})} \approx 1 + 2\sigma_{\max}(\mathbf{E}) = 1 + 2\|\mathbf{E}\|_2. \tag{24}$$

This completes the proof of the inequality chain presented in the main text.

## C  Experiment details

### C.1  Implementation of Traditional Baselines

**Traditional Baselines** We implement all the preconditioned conjugate gradient solvers in C++ code with reduced function call overhead, and use `nanobind` [35] to generate its Python binding. Even one method can have different implementations on different devices, leading to different iteration counts.

Table 7: Implementation of baseline preconditioners

| Device | Diag | IC | SPAI | AMG |
|---|---|---|---|---|
| CPU | Eigen [36] | Eigen [36] | cusplibrary [32] | PyAMG [37] |
| GPU | Custom | cuSPARSE | cusplibrary [32] | AMGX [38] |

All evaluations are performed on an AMD Ryzen 5 5600 CPU and an NVIDIA GeForce RTX 3060 GPU. The CPU frequency is fixed at 4.0 GHz and the OpenMP thread count is set to 4. The source code is compiled using GCC 14.2 and CUDA 12.8.

### C.2  Dataset Configurations

**Heat Equation/Poisson Equation**: The equations are discretized using the finite element method. We generate 9,147 samples on tetrahedral meshes from the TetWild dataset [28], employing P1 spatial discretization. The input node features of the GNNs are the nodes' position and the density.

**Hyperelasticity**  : We adopt standard optimization based time integration with stable Neo-Hookean material model [27], $\nu = 0.4$, and density $\rho = 1.0$. The geometry of the beam is defined as $H \times W \times L = 1 \times 1 \times 4$, with Dirichlet boundary conditions applied to the left and right boundaries. All physical parameters are specified in SI units. The domain is discretized at varying resolutions using TetGen [39]. The input node features are the nodes' position and one-hot vector to indicate whether the node is on the Dirichlet boundary.

**Synthetic**  : For each sample, we first construct $\mathbf{P}$ with a specified sparsity of $3 \times 10^{-4}$. To enforce the SPD property of $\mathbf{A}$, we compute $\mathbf{A} = \mathbf{P}\mathbf{P}^\top + \varepsilon\mathbf{I}$, where $\varepsilon =$1e-4 is chosen for numerical stability. The resulting matrix $\mathbf{A}$ has sparsity of 1.2e-3 approximately. The matrix size of $\mathbf{P}$ is randomly selected between 12,000 and 24,000. The input node feature $i$ is the $i$-th row average of matrix $\mathbf{A}$.

## D  Additional Experiments

### D.1  Performance on CPUs

Table 8: Benchmark results across different datasets. Total time $T_{\text{total}}$ (ms) and total iterations $k$ (in parentheses) until relative residual norm is less than $\text{rtol} = 10^{-8}$ are listed in the table. Lower values indicate better performance. The lowest value is in bold, and the second lowest is underlined.

| Device | Test Case | Diag | IC | AINV | Ours |
|---|---|---|---|---|---|
| GPU | Heat Equation | 77(520) | 167(204) | 78(330) | **36**(197) |
| | Poisson Equation | 45(320) | 101(128) | 58(217) | **26**(128) |
| | Hyperelasticity | 86(464) | 202(117) | 247(266) | **51**(175) |
| | Synthetic System | 445(2775) | 1399(1808) | 5024(10896) | **253**(1122) |
| CPU | Heat Equation | **83**(511) | 96(205) | 132(328) | 90(197) |
| | Poisson Equation | **53**(331) | 68(135) | 94(208) | 59(128) |
| | Hyperelasticity | **287**(464) | 299(160) | 453(266) | 361(175) |
| | Synthetic System | **950**(2776) | 1064(1221) | 14377(10906) | 1251(1122) |

## D.2 Comparison to AMG

**Algebraic Multigrid (AMG)** is a powerful tool for solving large-scale linear systems on both CPU and GPU architectures by exploiting the hierarchical structure of graph nodes. AMG can further serve as a preconditioner for CG solvers. While our method adopts a significantly different approach compared to AMG, we also include a comparison between AMG and our method on our datasets.

**Benchmark Results**: Our evaluation employs the default configurations listed in AMGX and PyAMG library for the evaluation across all datasets, while more precise settings could be applied to further improve its performance. As listed in Table 9, while effective for Heat/Poisson Equation problems, AMG can degrade the performance of CG solvers compared to other baseline preconditioners in more complex scenarios, due to challenges in selecting optimal parameters for smoothers and cycles.

Table 9: Benchmark result across different datasets. Total time $T_{\text{total}}$ (ms) and total iterations $k$ (in parentheses) until relative residual norm is less than $\text{rtol} = 10^{-8}$ are listed in the table. We examine two configurations: (1) AMG-preconditioned CG (AMG+CG) and (2) standalone AMG. "/" indicates that the default settings does not to converge within 10 seconds.

| Case | CPU | | | GPU | | |
| | AMG+CG | AMG | Ours | AMG+CG | AMG | Ours |
|---|---|---|---|---|---|---|
| Heat | 47(17) | 117(69) | 90(197) | 20(27) | **17**(18) | 36(197) |
| Heat-Large | 491(22) | 1312(103) | 1086(409) | **41**(10) | 94(17) | 154(409) |
| Poisson | 37(14) | 67(42) | 59(128) | **10**(9) | 38(30) | 26(128) |
| Hyperelasticity | 785(91) | / | 361(175) | 286(128) | / | **51**(175) |
| Synthetic | 6970(551) | / | 1251(1122) | / | / | **253**(1122) |

## D.3 Ablation Study on Matrix Norms

To validate the design choice of the custom matrix norm within our SAI loss, we conduct a targeted ablation study. As discussed in Section 3.3, our proposed mean norm was selected for its dimension-agnosticism, robustness to outliers, and low computational cost, offering key advantages over conventional norms such as the Frobenius and L1 norms. In this experiment, we evaluate the impact of this choice on the Heat dataset. We replace our norm with several standard alternatives while keeping all other model components and hyperparameters identical. The results are summarized in Table 10. They clearly demonstrate the superiority of our proposed norm. This represents a significant improvement over the Frobenius norm and the L1 norm. This quantitative evidence confirms that the specific properties of our chosen norm are not merely theoretical advantages but translate directly into tangible performance gains in the optimization process.

Table 10: Ablation study on the choice of matrix norm within the SAI loss. We report the average PCG iterations required for convergence on the "Heat" dataset.

| Norm Type | Iterations |
|---|---|
| Frobenius Norm ($\|\cdot\|_F$) | 222 |
| L1 Norm ($\|\cdot\|_1$) | 231 |
| **Ours** | **197** |

## D.4 Sensitivity Analysis of the Hyper-parameter $\varepsilon$

This section addresses the sensitivity of our method's performance to the value of the hyper-parameter $\varepsilon$ in (2). To evaluate this, we conducted an ablation study on the Heat dataset by using different values of $\varepsilon$. Table 11 summarizes the validation performance (measured in iterations) for the tested $\varepsilon$ values. The results demonstrate that our method is robust, with stable performance across three orders of magnitude ($3 \times 10^{-4}$ to $3 \times 10^{-2}$), indicating that $\varepsilon$ requires no careful per-instance tuning. However, an excessively large value of $\varepsilon$ (e.g., $3 \times 10^{-1}$) leads to training failure. We hypothesize that such a strong regularization term over-constrains the preconditioner, causing it to deviate excessively from the intended structure and thus degrading performance.

Table 11: Sensitivity analysis of the hyper-parameter $\varepsilon$ on the Heat dataset. Performance is measured by the number of iterations required on the validation set.

| $\varepsilon = 3 \times 10^{-5}$ | $\varepsilon = 3 \times 10^{-4}$ | $\varepsilon = 3 \times 10^{-3}$ | $\varepsilon = 3 \times 10^{-2}$ | $\varepsilon = 3 \times 10^{-1}$ |
|---|---|---|---|---|
| 222 | 208 | 197 | 205 | Training Failure |

### D.5 Generalization Analysis Against Learning-Based Baselines

This section provides a detailed analysis to address the generalization capabilities of our method compared to other learning-based approaches.

**Comparison with Neural PCG** For this experiment, we follow the OOD setting from Table 5 in the main paper and compare against Neural PCG [4]. Specifically, we decrease the density in the heat problem to an unseen value. Table 12 presents the total time (ms) and iteration counts (in parentheses) for both in-distribution and OOD settings. Our method not only outperforms Neural PCG in the in-distribution setting but also demonstrates significantly better generalization. While all methods experience performance degradation in the OOD setting, our method remains the most efficient.

Table 12: OOD generalization comparison with Neural PCG [4]. Total time (ms) and total iterations $k$ (in parentheses) are reported. The best and second-best results are bolded and underlined, respectively.

| Setting | IC | Diagonal | AINV | Neural PCG [4] | Ours |
|---|---|---|---|---|---|
| In-Distribution | 45(87) | 29(208) | 30(108) | 51(108) | **26(102)** |
| Out-of-Distribution | 337(876) | 262(1956) | 179(1039) | 268(743) | **167(1022)** |

**Comparison with Neural IF** We further test generalization against Neural IF [5] by increasing the sparsity of the synthesized matrix from $10^{-3}$ to $2 \times 10^{-3}$. The results are summarized in Table 13. Again, our method exhibits superior performance and robustness. In the more challenging OOD setting, our approach achieves a remarkable speedup over Neural IF, confirming its strong generalization capability.

Table 13: OOD generalization comparison with Neural IF [5]. The best and second-best results are bolded and underlined, respectively.

| Setting | IC | Diagonal | AINV | Neural IF [5] | Ours |
|---|---|---|---|---|---|
| In-Distribution | 440(385) | 166(970) | 2456(753) | 1040(354) | **132**(456) |
| Out-of-Distribution | 616(414) | 207(1040) | 3220(804) | 1222(375) | **168**(470) |

