# OpenReview forum: "Learning Sparse Approximate Inverse Preconditioners for Conjugate Gradient Solvers on GPUs"
_NeurIPS.cc/2025/Conference — NeurIPS 2025 poster_

### Official Review · Reviewer_umir · 2025-06-30

**Clarity:** 4
**Significance:** 3
**Originality:** 3
**Rating:** 5
**Confidence:** 4

**Summary:**

A new type of sparse approximate inverse is presented, based on GNNs, which incorporates a scaled minimization of the inverse in the Frobenius norm.  A stochastic estimator is used to reduce the cost of evaluating in this norm.  The results shows both fewer iterations in PCG and lower times in general for a suite of problems ranging from the heat equation to hyperelasticity to random (SPD) matrices.

**Questions:**

The paper identifies locality in this SPAI approach as being well-suited to the GPU, arguing two hops in G G^T.  Could the discussion be expanded?  Is performance (of the SpMV) better on the GPU due to density?

What happens for very large matrices?  Is the main limitation using a single GPU or are there other limitations?  For example if training on sizes from 400 -- 32k, can the model be used on matrices of size 1M?

**Ethical Concerns:**

["NO or VERY MINOR ethics concerns only"]

**Final Justification:**

Overall this work advances the field in a meaningful way.  There have been many attempts at advancing solvers and this paper make a clear contribution with the SPAI-based loss.  The results are solid and the paper is clear.  The authors did well with the rebuttals all around and mostly answered the questions.  I rated accept for the clarity and for the clear/meaningful contributions.

**Limitations:**

yes

**Quality:**

3

**Strengths And Weaknesses:**

The paper has several strengths.  It clearly motivates issues with previous GNN-based solvers and classical SPAI solvers.  The paper is also concise, very quickly summarizing the details of the method, the model, and the experimental setup.  Overall the results are quite good as well.  The appendix results on AMG are a good baseline (and address a predictable question).  Of course AMG will out perform for the Poisson problem here, but the reference point is very helpful.  The paper also compares with prior methods, another helpful data point.

One weakness is the size of the system.  Which appear to be limited to fairly small sizes at this point.  Also the train/test splitting of 4:1 raise the question of whether this is a practical solver in practice.

---

> ### Author Rebuttal · Authors · 2025-07-30
>
> We thank Reviewer umir for the encouraging review and for the questions that allow us to elaborate on the core strengths of our method.
>
> > **Concern 1:** The paper identifies locality in this SPAI approach as being well-suited to the GPU, arguing two hops in $G G^T$. Could the discussion be expanded?
>
> Our method's suitability for GPUs stems from two synergistic aspects:
>
> 1. **Computational Locality:** The preconditioner application ($M^{-1}r=GG^T r$) relies on sparse matrix-vector multiplication (SpMV), a highly parallelizable operation that is heavily optimized in modern GPU libraries.
> 2. **Learning Locality:** More fundamentally, the approximation $A^{-1} \approx GG^T$ is itself local (see Equation (3) in our paper). This aligns perfectly with the local message-passing mechanism of GNNs, allowing the model to efficiently and effectively learn these structural patterns.
>
> > **Concern 2:** What happens for very large matrices? Is the main limitation using a single GPU or are there other limitations? For example if training on sizes from 400 -- 32k, can the model be used on matrices of size 1M?
>
> This is an excellent question about the method's scalability.
>
> - **Inference Stage:** The computational complexity during inference scales near-linearly with the number of non-zero entries (nnz) under fixed GNN architecture (e.g., layers/hidden dimensions). This suggests that applying the trained model to million-scale sparse matrices is potentially feasible, provided that both the model and matrix data can fit within GPU memory. Our largest validated case with matrices up to ~100k ("Heat-Large" in Table 5) shows stable performance without retraining. However, we acknowledge that for extremely large or structurally complex matrices, deeper message-passing or adaptive architectures might be needed — which could affect scalability.
> - **Training Stage:** The primary limitation is in training, where GPU memory for holding very large graphs becomes a bottleneck. This is a general challenge in the GNN field, not specific to our method. To contextualize scale: recent GNN works handle graphs with 50k–2M nodes [1], 14k–1.5M nodes [2], and 89k–2.4M nodes [3] (the number of nodes corresponds to the size of matrix in this work). We believe training scalability could potentially be addressed through established GNN techniques (e.g., graph sampling, distributed training), though this is beyond our current scope and would require dedicated validation.
>
> [1] Wei-Lin Chiang, Xuanqing Liu, Si Si, Yang Li, Samy Bengio, and Cho-Jui Hsieh. 2019. Cluster-GCN: An Efficient Algorithm for Training Deep and Large Graph Convolutional Networks. In Proceedings of the 25th ACM SIGKDD International Conference on Knowledge Discovery & Data Mining (KDD '19).
>
> [2] Hanqing Zeng, Hongkuan Zhou, Ajitesh Srivastava, Rajgopal Kannan and Viktor Prasanna. 2020. GraphSAINT: Graph Sampling Based Inductive Learning Method. International Conference on Learning Representations
>
> [3] Rui Xue, Haoyu Han, MohamadAli Torkamani, Jian Pei, and Xiaorui Liu. 2023. LazyGNN: large-scale graph neural networks via lazy propagation. In Proceedings of the 40th International Conference on Machine Learning (ICML'23)

---

> > ### Comment · Reviewer_umir · 2025-08-06
> >
> > Thank you for the detail responses here and to the other reviews -- I enjoyed the paper and found all of the follow-ups helpful.
> >
> > I'm still a bit confused by the SpMV on the GPU.  In general a SpMV will be dependent on both format and the sparsity pattern (both contributing to efficient memory coalescing).  Equation 3 shows that the preconditioner inherits the sparsity of A.  It's an interesting point that it fits well with GNN architectures, but it's still unclear why the operations should be (more) efficient on a GPU.
> >
> > Thank you for the details on scalability.  It may be helpful to spell that out more clearly.  On one hand the sizes are unremarkable, up to 32k or so (or even 100k), but considering a single GPU makes it more relevant, especially if you could speculate on how a multi-GPU solver might be envisioned.
> >
> > Anyway, I'm holding on to my original (good) rating despite the lingering questions.

---

> > > ### Author Response · Authors · 2025-08-08
> > >
> > > Thank you for the thoughtful follow-up and your continued engagement with our work. We appreciate you raising these important points, which will help us improve the clarity of the paper.
> > >
> > > Regarding SpMV performance, our primary contribution is at the algorithmic level, showing that GNNs can effectively learn local preconditioners. We agree that translating this into optimal hardware performance requires system-level tuning. In our current work, we used standard CSR format without further optimization. Our experimental matrices—obtained from standard finite‑element discretization of PDEs—generally exhibit relatively uniform sparsity patterns, which may partially explain the favorable GPU performance observed, though this deserves more systematic analysis. We fully agree that a deeper investigation into how our method's learned sparsity patterns interact with different storage formats and memory access patterns is a crucial and valuable direction for future work, which we will discuss more clearly in the paper.
> > >
> > > Regarding scalability, thank you for the valuable suggestion. We will expand our discussion to more clearly delineate the memory limitations of a single GPU and outline a potential path towards a multi-GPU solver. As you suggest, this could leverage established techniques from the distributed GNN literature [1,2,3], and we will add references to this promising line of future work.
> > >
> > > We are grateful for this insightful feedback, which will certainly strengthen the paper. We appreciate your positive assessment and support.
> > >
> > > [1] Yunyong Ko, Kibong Choi, Jiwon Seo, and Sang Wook Kim. 2021. An in-depth analysis of distributed training of deep neural networks. In Proceedings of the IEEE International Parallel and Distributed Processing Symposium. 994–1003.
> > >
> > > [2] Matthias Langer, Zhen He, Wenny Rahayu, and Yanbo Xue. 2020. Distributed Training of Deep Learning Models: A
> > > Taxonomic Perspective. IEEE Transactions on Parallel and Distributed Systems 31, 12 (2020), 2802–2818.
> > >
> > > [3] Shao, Y., Li, H., Gu, X., Yin, H., Li, Y., Miao, X., Zhang, W., Cui, B., & Chen, L. (2022). Distributed Graph Neural Network Training: A Survey. ACM Computing Surveys, 56, 1 - 39.

---

### Official Review · Reviewer_GmV6 · 2025-07-01

**Clarity:** 3
**Significance:** 3
**Originality:** 3
**Rating:** 5
**Confidence:** 3

**Summary:**

This paper proposes a GPU-friendly preprocessing technique based on a graph neural network (GNN) that learns a Sparse Approximate Inverse. Most existing approaches rely on Cholesky factorization, whose triangular solves become bottlenecks on GPUs when used with the Conjugate Gradient (CG) method. To eliminate this bottleneck, the authors directly construct an approximation $A^{-1} \approx GG^\top$ with a GNN and use it as the preconditioner, reducing the it to sparse matrix–vector multiplications that run efficiently on GPUs. They also devise a scale-invariant loss function for training the GNN, which further improves performance. Numerical experiments on several PDE problems and on synthetic data demonstrate that the proposed method outperforms.

**Questions:**

1. The authors claim strong generalization, and Table 5 indeed shows good performance on out-of-distribution problems. However, is this superiority also observed relative to other learning-based methods? For example, do the baselines in Table 4 fail to generalize under the conditions of Table 5?
2. If the proposed method does generalize better than competing approaches, what is the underlying reason? Could the locality of the computations be a factor?
3. The parameter $\varepsilon$ in $GG^\top + \varepsilon I$ is a hyper-parameter. How sensitive is the method’s performance to its value?

**Ethical Concerns:**

["NO or VERY MINOR ethics concerns only"]

**Final Justification:**

Thank you for your reply. I will keep my score.

**Limitations:**

yes

**Paper Formatting Concerns:**

I was unable to print this paper, possibly due to a font-related issue.

**Quality:**

4

**Strengths And Weaknesses:**

## Strengths
- The manuscript is well organized and clearly written, making it very easy to follow.
- Algorithm design that takes hardware characteristics into account is important; given the rapid performance gains of modern GPUs, GPU-oriented preconditioners such as the one proposed here are valuable.
- The paper identifies triangular solves in factorization-based preconditioners as a key bottleneck and offers a simple yet effective way to avoid them.
- The numerical experiments are comprehensive and thoughtfully designed; the purpose of each experiment is clear. Reporting both iteration counts and wall-clock times gives a full picture of the algorithm’s behavior. The per-stage timing analysis in Table 3 is particularly informative.

## Weaknesses
No major shortcomings were found; if anything:
- Figures are generally small and hard to read.
- The method offers no special advantage in CPU computations (though this is arguably outside the scope of the work).

---

> ### Author Rebuttal · Authors · 2025-07-30
>
> We are grateful to Reviewer GmV6 for the positive feedback and for asking insightful questions that help us strengthen our claims. The figures will be clarified in the final version to ensure readability.
>
> > **Concern 1 & 2:** The authors claim strong generalization, and Table 5 indeed shows good performance on out-of-distribution problems. However, is this superiority also observed relative to other learning-based methods? For example, do the baselines in Table 4 fail to generalize under the conditions of Table 5?
> >
> > If the proposed method does generalize better than competing approaches, what is the underlying reason? Could the locality of the computations be a factor?
>
> We appreciate this excellent question. We compare the generalization capability of each learning-based methods on the same dataset shown in Table (4) following Table 5's OOD settings. Total time (ms) and total iterations k (in parentheses) are listed in the tables below.
>
> **vs. Neural PCG [1]:** we decrease the density of mesh in the heat problem to an unseen value.
>
> |                     | IC       | Diagonal  | AINV      | Neural PCG[1] | Ours          |
> | ------------------- | -------- | --------- | --------- | ------------- | ------------- |
> | In distribution     | 45(87)   | 29(208)   | 30(108)   | 51(108)       | **26**(102)   |
> | Out of distribution | 337(876) | 262(1956) | 179(1039) | 268(743)      | **167**(1022) |
>
> **vs. Neural IF [2]:** we increase the sparsity of the synthesized matrix from 1e-3 to 2e-3.
>
> |                     | IC        | Diagonal  | AINV      | Neural IF[2] | Ours         |
> | ------------------- | --------- | --------- | --------- | ------------ | ------------ |
> | In distribution     | 440(385)  | 166(970)  | 2456(753) | 1040(354)    | **132**(456) |
> | Out-of-distribution | 616 (414) | 207(1040) | 3220(804) | 1222(375)    | **168**(470) |
>
> - In distribution: our method is 96% faster compared with Neural PCG[1] and 687% faster compared with Neural IF[2];
> - Out of distribution: our method is 60% faster compared with Neural PCG[1] and 627% faster compared with Neural IF[2].
>
> We strongly concur with your intuition that **locality** is fundamental to our method's strong generalization. Our preconditioner inherently captures local, transferable physical patterns from the graph structure. This property enables robustness across diverse scales and geometries, unlike global methods (e.g., IC-based) that tend to overfit to training data and struggle with distribution shifts.
>
> > **Concern 3:** The parameter $\varepsilon$ in $GG^\top + \varepsilon I$ is a hyper-parameter. How sensitive is the method’s performance to its value?
>
> Thank you for asking about this important hyper-parameter. We conducted an ablation study to evaluate the sensitivity to $\varepsilon$, confirming that our method is robust and does not require careful per-instance tuning. We trained different models from scratch using various $\varepsilon$ values on the "Heat" dataset. The validation performance (in iterations) confirms that the model is not sensitive to the precise choice of $\varepsilon$ within a reasonable range.
>
> | $\varepsilon$ | 3e-5 | 3e-4 | 3e-3    | 3e-2 | 3e-1 |
> | :-------------- | :--- | :--- | :------ | :--- | :--- |
> | Val. Iterations | 222  | 208  | **197** | 205  | Training Failure |
>
> Performance is stable and efficient for $\varepsilon$ between 3e-4 and 3e-2. Too large a value (1e-1) may over-regularize, causing the preconditioner to deviate from $A^{-1}$, which leads to performance degradation. This analysis robustly supports our claim that $\varepsilon$ generally does not require instance-specific fine-tuning within a practical range.

---

> > ### Comment · Reviewer_GmV6 · 2025-08-07
> >
> > I thank the authors for the detailed explanation and the additional experiments. I am pleased with the clarifications and confirm that my overall score remains unchanged.

---

### Official Review · Reviewer_JerM · 2025-07-03

**Clarity:** 3
**Significance:** 2
**Originality:** 2
**Rating:** 4
**Confidence:** 4

**Summary:**

This paper addresses the challenge of accelerating the Conjugate Gradient (CG) solver for symmetric positive definite linear systems on GPUs. The paper proposes a method that use GNNs to construct Sparse Approximate Inverse (SPAI) preconditioners. This method is suitable for GPU because required only matrix-vector products which are highly parallelizable. The results demonstrate that the proposed method outperforms standard traditional and previous learning-based preconditioners on GPUs.

**Questions:**

* The **SAI loss** is a key contribution. It uses a non-standard definition for the matrix norm: $\Vert A \Vert = mean_{A_{ij} \not= 0} \vert A_{ij}\vert$.
	* Could authors explain the motivation for choosing this specific heuristic instead of more conventional matrix norms (e.g., the Frobenius, spectral, or L1 norm)?
	* How does the performance of this specific matrix form compare to that of conventional matrix norms, such as the Frobenius, spectral, or L1 norm?
* Could authors give the results of GPU-to-GPU comparison against other learning-based baselines?
* Could authors  consider heat equation with $a(x)$ with high contrast values?
* Could authors give results  of the method's performance on non-smooth PDE fields?

**Ethical Concerns:**

["NO or VERY MINOR ethics concerns only"]

**Final Justification:**

Thank you for your reply. My concern have been resolved

**Limitations:**

The authors discussed the method limitations in the conclusion.

**Quality:**

3

**Strengths And Weaknesses:**

**Strengths**
* The paper is well written. I appreciated the smooth introduction on the preconditioning of linear systems, as well as the mathematical rigor throughout the paper.
* The method is tested on three different PDE-derived problems (Heat, Poisson, Hyperelasticity) and a synthetic dataset.
* The proposed method is compared against a comprehensive set of baselines, including standard traditional methods (Diag, IC, AINV) and recent learning-based approaches ([1], [2]).
* The author provides results for both CPU and GPU across different datasets.
* The author provides a comparison of the proposed loss function $L_{\text{SAI}}$ to the cosine similarity loss $L_{\text{CS}}$ and scale-variant loss $L_{2}$.

**Weaknesses**
* This loss function $L_{\text{SAI}}$ was previously introduced in article [1], with a detailed investigation and theoretical justification provided in [3].
* In Table 4, the GPU performance for the learning-based baselines [1, 2] is marked as "N/A" (or "/"). The authors justify this by stating that a component they use IC is slow on a GPU. While this justification is reasonable and supports their main idea, it also means there is no direct GPU-to-GPU comparison against other learning-based baselines. Consequently, the GPU speedup is demonstrated only against traditional methods.
* The authors consider heat equation with $a(x)$ (randomly sampled on the mesh, with values ranging from $1 \cdot 10^{-4}$ to $5 \cdot 10^{-4}$). The proposed method seems to be mostly tested on very smooth equations.


[1] -- Li, Y., Chen, P. Y., Du, T., & Matusik, W. (2023, July). Learning preconditioners for conjugate gradient PDE solvers. In International Conference on Machine Learning (pp. 19425-19439). PMLR.

[2] -- Häusner, P., Öktem, O., & Sjölund, J. (2023). Neural incomplete factorization: learning preconditioners for the conjugate gradient method. arXiv preprint arXiv:2305.16368.

[3] -- Trifonov, V., Rudikov, A., Iliev, O., Laevsky, Y. M., Oseledets, I., & Muravleva, E. (2024). Learning from linear algebra: A graph neural network approach to preconditioner design for conjugate gradient solvers. arXiv preprint arXiv:2405.15557.

---

> ### Author Rebuttal · Authors · 2025-07-30
>
> We sincerely thank Reviewer JerM for the detailed feedback and for raising several critical questions. Your comments have helped us significantly improve the clarity and completeness of our work.
>
> > **Concern 1:** This loss function was previously introduced in article [1], with a detailed investigation and theoretical justification provided in [3].
>
> We would like to clarify the fundamental difference between our work and references [1, 3]. Those methods aim to learn a Cholesky decomposition of the **original matrix ($A \approx LL^T$)**. In contrast, our goal is to directly learn a decomposition of the **inverse matrix ($A^{-1} \approx GG^T$)**. This is a fundamentally different objective, which necessitates our novel SAI loss function designed specifically for this purpose. We will emphasize this distinction more clearly in the revised manuscript.
>
> > **Concern 2:** The SAI loss is a key contribution. It uses a non-standard definition for the matrix norm.
> >
> > - Could authors explain the motivation for choosing this specific heuristic instead of more conventional matrix norms?
> > - How does the performance of this specific matrix form compare to that of conventional matrix norms?
>
> We chose this specific norm ($\Vert A \Vert = \mathrm{mean}_{A_{ij} \not= 0} \vert A_{ij}\vert$) for three main reasons:
>
> 1. **Dimension-Agnostic:** Unlike the Frobenius norm (where $\|I_n\|_F = \sqrt{n}$), our norm is insensitive to matrix size, which is crucial for training on datasets with varying matrix sizes.
> 2. **Robustness:** Compared to $L_1$ norms ($\|A\|_1 = \max_{1 \leq j \leq n} \sum_{i=1}^{m} |a_{ij}|$), which sum absolute values and can be dominated by sparse but large entries (common in ill-conditioned matrices), our averaging approach is less sensitive to individual entries, leading to more robust performance.
> 3. **Simplicity**: The spectral norm, defined as the largest singular value of $A$, is costly to compute, while our norm is a simple and inexpensive measure of average nonzero magnitude.
>
> To quantitatively validate our choice, we conducted an ablation study on the “Heat” dataset. We trained our model with different loss norms and compared the average PCG iterations required for convergence. The results clearly demonstrate the superiority of our proposed norm.
>
> | Loss Norm       | Avg. Iterations |
> | :-------------- | :-------------- |
> | Frobenius Norm  | 222             |
> | L1 Norm         | 231             |
> | **Mean (Ours)** | **197**         |
>
> > **Concern 3:** Could authors give the results of GPU-to-GPU comparison against other learning-based baselines?
>
> We thank the reviewer for this suggestion. We conducted the requested GPU-to-GPU comparison on the respective datasets of the baselines. The results clearly demonstrate the significant computational advantage of our method.
>
> - **vs. Neural PCG [1]:**
>   - Neural PCG: 51 ms (108 iterations)
>   - Ours: **26 ms** (102 iterations) (1.96x speedup)
> - **vs. Neural IF [2]:**
>   - Neural IF:  1040 ms (354 iterations)
>   - Ours: **132 ms** (456 iterations) (7.88x speedup)
>
> > **Concern 4:** Could authors consider heat equation with high contrast values? Could authors give results of the method's performance on non-smooth PDE fields?
>
> Thank you for this insightful suggestion. We designed a heat problem with piecewise constant density values to demonstrate performance on non-smooth, high-contrast PDE problems. We set the higher density to 1e-2 and lower density to 1e-4 (100× contrast ratio), while keeping other parameters and meshes unchanged. We used the model trained on the “Heat” dataset described in the paper, and directly applied the preconditioner on this high-contrast dataset.
>
> | Preconditioner      | Time (ms) | Iterations |
> | ------------------- | --------- | ---------- |
> | Diagonal            | 60        | 460        |
> | Incomplete Cholesky | 121       | **159**    |
> | AINV                | 65        | 292        |
> | **Ours**            | **40**    | 184        |
>
> Notably, our method achieves the **fastest solution time**, substantially outperforming all baselines despite the discontinuous coefficients. Although Incomplete Cholesky requires fewer iterations, its higher per-iteration computational cost results in slower overall performance. These results validate that our method effectively handles high-contrast, non-smooth PDE fields without requiring specialized modifications.
>
> [1] Li, Y., Chen, P. Y., Du, T., & Matusik, W. (2023, July). Learning preconditioners for conjugate gradient PDE solvers. In International Conference on Machine Learning (pp. 19425-19439). PMLR.
>
> [2] Häusner, P., Öktem, O., & Sjölund, J. (2023). Neural incomplete factorization: learning preconditioners for the conjugate gradient method. arXiv preprint arXiv:2305.16368.
>
> [3] Trifonov, V., Rudikov, A., Iliev, O., Laevsky, Y. M., Oseledets, I., & Muravleva, E. (2024). Learning from linear algebra: A graph neural network approach to preconditioner design for conjugate gradient solvers. arXiv preprint arXiv:2405.15557.

---

> > ### Comment · Reviewer_JerM · 2025-08-05
> >
> > Thank you for your comprehensive response. I appreciate the answers to my questions. I  raised my score.

---

### Official Review · Reviewer_Rbu2 · 2025-07-03

**Clarity:** 2
**Significance:** 3
**Originality:** 3
**Rating:** 4
**Confidence:** 3

**Summary:**

This paper aims to solve the linear system with sparse weights via the proposed
precondition conjugate gradient (CG) method. The linear system focuses on the graph
field applications, and this precondition CG method utilizes a learnable graph neural
network’s (GNN’s) output as the preconditioner.

**Questions:**

See "weaknesses"

**Ethical Concerns:**

["NO or VERY MINOR ethics concerns only"]

**Final Justification:**

An interesting direction though the eventual practical use may still have some way to go.

**Limitations:**

yes

**Quality:**

3

**Strengths And Weaknesses:**

#### Strengths

- Problem setup is clear.
- The author designed a learning-based presconditioner that can utilize GPU for the CG computing.

#### Weaknesses

- The approach of using GNN to generate apreconditioner may not be very general. For each type of applications, it seems we must train a GNN, a process that is time consuming. For example, for the four types of applications considered in Section 4, we must rain a GNN for each of them. Further, for the same type of problems, the GNN training is uder some fixed settings (e.g., 9,147 meshes in line 210). The effectiveness of the proposed methods may depend on these settings.

- On GPU, while the proposed method is the fastest, the simplest diagonal preconditioner works quite well. So in the practical use, will people switch to the proposed method?

- It is useful to discuss the training time for GNN models.
- In line 86, SPAI stands for Sparse Approximate Inverse.
  However, in line 237, the authors switch to use AINV.

---

> ### Author Rebuttal · Authors · 2025-07-30
>
> We thank Reviewer Rbu2 for the constructive feedback and positive evaluation of our work.
>
> > **Concern 1:** The approach of using GNN to generate a preconditioner may not be very general. For each type of application, it seems we must train a GNN, a process that is time consuming. Further, for the same type of problems, the GNN training is under some fixed settings. The effectiveness of the proposed methods may depend on these settings.
> >
> > It is useful to discuss the training time for GNN models.
>
> Our model, once trained offline, can be directly applied to solve the same PDE under various boundary conditions, physical parameters, and meshes. This "train once, use many times" paradigm offers significant generality.
>
> Compared to traditional methods like AMG, which often require expert-level manual parameter tuning for each new problem instance, our GNN-based approach automates this process. The GNN learns to generate efficient preconditioners directly from data, representing a more general paradigm.
>
>
> Regarding the training time, we provide the following details for our one-time offline training process:
>
> - **Heat & Poisson:** ~24 hours on a single A100 GPU.
> - **Elasticity:** ~10 hours on a single RTX 3060 GPU.
> - **Synthetic:** ~12.2 hours on a single A100 GPU.
>
> > **Concern 2:** On GPU, while the proposed method is the fastest, the simplest diagonal preconditioner works quite well. So in the practical use, will people switch to the proposed method?
>
> We appreciate the reviewer raising this practical point. We fully acknowledge that diagonal preconditioners (e.g., Jacobi) offer significant advantages in simplicity, minimal implementation overhead, and efficient GPU parallelization, making them a strong and widely adopted baseline for many real-world applications—especially on smaller or less complex problems.
>
> However, for large-scale and highly challenging problems, our method provides substantial practical gains. It achieves a 2x speedup over the diagonal preconditioner on standard benchmarks (e.g., Table 3). Crucially, this gap widens with problem complexity: on the “Heat-Large” dataset (Table 5), our method **saves ~100ms** per solve, and in the extreme heat equation experiment (density=1e-5, as detailed in our response to Reviewer JerM), it solves the system in **29 ms** versus **56 ms** for the diagonal preconditioner. In contexts where milliseconds matter—such as real-time simulation or billion-scale systems -- this reduction translates to meaningful efficiency improvements.
>
> Therefore, while diagonal preconditioners remain ideal for lightweight tasks, our method offers a valuable alternative for demanding scenarios where computational speed is critical, justifying its adoption in such cases.
>
> > **Concern 3:** In line 86, SPAI stands for Sparse Approximate Inverse. However, in line 237, the authors switch to use AINV.
>
> Thank you for pointing this out. Sparse Approximate Inverse (SPAI) is a class of methods, aiming at using a sparse matrix to approximate the $A^{-1}$. AINV is a specific, widely-used implementation of SPAI that uses a factorization to guarantee the positive definiteness of the preconditioner, i.e., $A^{-1}\approx G G^T$, where $G$ is sparse [1]. We will clarify this distinction and add the appropriate citation in the final version of the paper.
>
> [1] J. Scott and M. Tůma, Algorithms for Sparse Linear Systems. in Nečas Center Series. Cham: Springer International Publishing, 2023. doi: 10.1007/978-3-031-25820-6.

---

> > ### Comment · Reviewer_Rbu2 · 2025-08-04
> >
> > Thank the authors for address my concerns. I believe that this topic is a good direction to explore, but from the long GNN training and the competitive performance of simple diagonal preconditioners, I guess there is still some way to go for this approach to be widely adopted

---

> > > ### Author Response · Authors · 2025-08-04
> > >
> > > We sincerely thank the reviewer for their insightful comments and for recognizing our approach's potential. We agree that practical considerations, such as initial training cost and simpler baselines' competitiveness, are crucial for broader adoption. These are important challenges we are committed to addressing by optimizing efficiency and expanding applicability. We are excited to continue exploring this promising direction.

---

### Decision · Program_Chairs · 2025-09-17

**Decision:**

Accept (poster)

**Comment:**

This paper studies the problem of finding effective preconditioners for solving symmetric positive definite linear systems. The main contribution of the work is a learning-based method that relies on graph neural networks to construct sparse approximate inverse preconditioners, and which is naturally suitable for computation on GPUs. All reviewers agreed that the paper is well written and clearly introduces the problem setting and how their approach fits with existing works, and Reviewers JerM and GmV6 both noted the comprehensive nature of the empirical results comparing to established baselines and reporting both iterations and wall-clock time. Overall, the reviewers were positive about the paper, and I agree with the consensus that it would be a worthy addition to the conference.